# Health-Related Quality of Life in Levothyroxine-Treated Hypothyroid Women and Women without Hypothyroidism: A Case–Control Study

**DOI:** 10.3390/jcm9123864

**Published:** 2020-11-27

**Authors:** Benjamín Romero-Gómez, Paula Guerrero-Alonso, Juan Manuel Carmona-Torres, Diana P. Pozuelo-Carrascosa, José Alberto Laredo-Aguilera, Ana Isabel Cobo-Cuenca

**Affiliations:** 1Hospital El Tomillar de Sevilla, Servicio Andaluz de Salud (SAS), 41500 Alcalá de Guadaira, Spain; volvart@hotmail.com; 2Centro de Salud Najera, Servicio Rioja Salud, 26300 Najera, Spain; pguerrero@riojasalud.es; 3Facultad de Fisioterapia y Enfermería de Toledo, Universidad de Castilla la Mancha, 45005 Toledo, Spain; juanmanuel.carmona@uclm.es (J.M.C.-T.); dianaP.Pozuelo@uclm.es (D.P.P.-C.); anaisabel.cobo@uclm.es (A.I.C.-C.); 4Grupo de Investigación Multidisciplinar en Cuidados (IMCU), Universidad de Castilla la Mancha, 45005 Toledo, Spain; 5Instituto Maimónides de Investigación Biomédica de Córdoba (IMIBIC-GE08), 14004 Córdoba, Spain

**Keywords:** hypothyroidism, quality of life, health, women, thyroid hormone

## Abstract

The use of levothyroxine is not always related to the elimination of the symptoms of hypothyroidism. The aim of this study is to compare the health-related quality of life (HRQOL) of a group of hypothyroid women under levothyroxine treatment with that of a group of non-hypothyroid women. Methodology: A case–control study was performed. We used convenience sampling. The case group consisted of 152 levothyroxine-treated hypothyroid women; the control group consisted of 238 women without hypothyroidism disorders. All of the participants were euthyroid according to the clinical practice guidelines. We used as instruments the Short Form-12 questionnaire (SF-12v1) and a sociodemographic questionnaire. Results: Hypothyroid women scored significantly lower in HRQOL in SF-12v1 mental and physical components than the control group (mental component summary: 41.23 ± 12.12 vs. 46.45 ± 10.22, *p <* 0.001; physical component summary: 49.64 ± 10.16 vs. 54.75 ± 5.76, *p <* 0.001). body mass index (BMI) and age showed an influence on the physical component (*p <* 0.001 in both variables). Adjusted for age and BMI, hypothyroidism was still related to worse scores (*p <* 0.001). Conclusion: Despite being euthyroid, women with hypothyroidism showed a poorer quality of life than women without hypothyroidism. Health professionals need to assess the HRQOL of women with hypothyroidism. Further research on HRQOL and hypothyroidism is needed.

## 1. Introduction

The thyroid gland plays an important role in the regulation of metabolism through the synthesis of two hormones—thyroxine (T4) and triiodothyronine (T3) [1]. These hormones have receptors in very different parts of the body, from peripheral tissues to brain structures [2,3]. This may explain why alterations in its secretion, either in excess or defect, can lead to diverse symptoms in the organism.

Among thyroid disorders, hypothyroidism is the most common, especially in women [4]. Hypothyroidism is defined by insufficient hormone production by the thyroid gland. In response, hypophysis increases the levels of the thyroid-stimulating hormone (TSH) in order to stimulate the thyroid gland. This compensatory mechanism is effective for a while, although it may become insufficient. When it occurs, the disease is called overt hypothyroidism. In this new situation, patients show low serum levels of thyroid hormones and an increase in TSH serum levels [5]. Hypothyroidism is a disease with an insidious course, characterized by very subtle initial symptoms that hinder the diagnosis. It may cause very different disorders: tiredness, impaired cognition, anxiety disorders, depressive symptoms, decreased information processing speed, reduced efficiency of executive functions, poor learning, dry and coarse skin, muscle weakness, sexual dysfunction, constipation, and weight gain [6,7,8,9,10,11].

While hypothyroidism may have different etiologies, the most common treatment is the intake of levothyroxine sodium (LT4) after an overnight fast [12]. It allows the recovery of T4 hormone serum levels [13] and, indirectly, the normalization of TSH levels. TSH levels are usually considered the indicator of treatment efficacy. The treatment is considered appropriate when TSH levels return to their normal range (0.4−4.0 mIU/L) [12]. Besides being an effective, safe, and simple treatment [5], the use of levothyroxine clearly reduces symptoms for most of the patients [8,14,15,16,17].

Nonetheless, it has been observed in recent years that symptoms remain in some patients despite reaching euthyroidism [5,18]. For example, the persistency of tiredness complaints, cognitive dysfunction, sexual disorders, mood disorders, working memory disorders, and motor learning disorders in properly treated patients have been reported [11,19,20,21,22]. On the contrary, several authors found no more neuropsychological problems in treated hypothyroid patients than in controls [23,24,25]. Similarly, Engum et al. reported no unusual anxiety or depression disorders in patients with thyroid disorders treated with levothyroxine [26].

The effects of levothyroxine treatment on health-related quality of life (HRQOL) are under research. HRQOL is a wide concept that implies the subjective assessment of different facets of life and their relationship with well-being [27]. Two main measurement components are usually differentiated: the physical component summary (PCS) and the mental component summary (MCS) [28,29]. Similar to other chronic diseases, nontreated hypothyroidism reduces the HRQOL of patients [22,30]. Furthermore, it has been proven that levothyroxine treatment improves the HRQOL of these patients [22]. However, this improvement seems incomplete; several authors have suggested that mean scores in HRQOL are lower in treated patients than in the general population [19,31]. On the other hand, Rakhshan et al. does not report those differences in treated patients [32].

Faced with such contradictory data, the aim of this study is to compare the health-related quality of life (HRQOL) of a group of hypothyroid women under levothyroxine treatment with that of a group of non-hypothyroid women.

## 2. Materials and Methods

### 2.1. Design and Study Population

From October 2018 to March 2019, 390 adult women (18–64 years old) were recruited among the patients of two primary care health zones in Spain. All of them were euthyroid. Women were divided into two groups: (1) the case group: 152 women with primary hypothyroidism treated with levothyroxine; (2) the control group: 238 women without thyroid disorders.

Inclusion criteria for the case group were (1) previous medical diagnosis of primary hypothyroidism, (2) at least 6 months of levothyroxine treatment, with the same dose, (3) normalized TSH levels (0.4–4.0 mIU/L). Exclusion criteria were (1) diagnosis of secondary or subclinical hypothyroidism, (2) abnormal TSH levels, (3) being pregnant or lactating, (4) diagnosis of psychiatric disease, endocrine disorder, or serious disease (e.g., cancer, heart problems), or (5) consumption of medications that may interfere with thyroid function (e.g., chemotherapy).

Inclusion criteria for the control group were (1) absence of thyroid disorders, and (2) euthyroidism (TSH levels 0.4–4.0 mIU/L). Exclusion criteria were (1) abnormal TSH levels, (2) being pregnant or lactating, (3) diagnosis of psychiatric disease, endocrine disorder, or serious disease (e.g., cancer, heart problems), and (4) consumption of medications that may interfere with thyroid function (e.g., chemotherapy).

### 2.2. Sample Size

Using the Short Form-12 questionnaire (SF-12v1), Vilagut et al. found a mean score for the physical component summary (PCS) of 52.13 ± 8.28 and for the mental component summary (MCS) of 50.58 ± 9.13 in Spanish women between 35 and 49 years old [33]. Based on these data, a sample of 59 women in each group was considered sufficient, with an α risk of 0.05, a β risk of 0.2, a level of confidence of 95%, and a replacement rate of 10%. Sample size was calculated using Granmo software (version 7.12, Antaviana, Institut Municipal d´investigatio medica, Barcelona, Spain).

### 2.3. Instruments

An online questionnaire was constructed. It included two instruments to measure the variables.

(1)Questionnaire on sociodemographic variables (age, tobacco consumption, educational level, employment status, civil status, cohabitation, and sexual orientation). Clinical variables—BMI, etiology of hypothyroidism, and TSH levels—were collected during the patients’ annual reviews.(2)SF-12v1 [34,35]. It was designed to measure HRQOL. It assesses levels of well-being and functional capacity in people aged 14 or older in two dimensions: PCS and MCS. It comprises 12 items with Likert-type and dichotomous responses. Scores are transformed and normalized in order to obtain the final result [36]. The mean score in the Spanish population is established at 50 ± 10 [33]. This mean is a validated and reliable measure, with a Cronbach’s alpha greater than 0.70 in both components [33].

### 2.4. Data Collection

Healthcare professionals from both primary care health zones were contacted and invited to participate in our study. 

Convenience sampling was used. Subjects were recruited when they attended a scheduled visit to follow-up on their thyroid disease. They all had to be fluent in Spanish. Moreover, inclusion and exclusion criteria were reviewed. After giving their informed consent, their height, weight, and results of the blood tests were collected. All blood samples were taken at 8.00 a.m. after an 8-h overnight fast. Only euthyroid women (TSH 0.4–4.0 mIU/L) were selected. Participants were not directly contacted by the researchers. Women who met all the criteria were provided with a link to the online questionnaire. This questionnaire was anonymous and did not collect any personal data other than the variables of this study.

### 2.5. Study Variables

#### 2.5.1. Independent Variables

Sociodemographic variables: age (quantitative), educational level (categorical), employment status (categorical), sexual orientation (categorical), civil status (categorical), and cohabitation (dichotomous). Clinical variables: presence/absence of hypothyroidism (dichotomous), etiology of hypothyroidism (categorical), tobacco consumption (dichotomous), and BMI (quantitative/categorical).

#### 2.5.2. Dependent Variables

The SF-12v1 questionnaire measures two different components: PCS and MCS. These variables were used as both quantitative and categorical variables.

### 2.6. Statistical Analysis

A descriptive analysis of the variables was performed by calculating counts (n) and proportions (%) of the qualitative variables and means (m) and standard deviations (SD) of the quantitative variables. The Kolmogorov–Smirnov test was used to test for data normality. Quantitative data were analyzed by Student’s *t*-test. A χ-square test was used to analyze the categorical variables. Finally, a univariable logistic regression model and an analysis of covariance (ANCOVA) adjusted for age and BMI were performed. All hypotheses contrasts were bilateral, and a value of *p <* 0.05 was considered significant in all tests. Data were analyzed using the statistical software IBM SPSS (version 22.0, IBM Corp, Armonk, NY, USA).

### 2.7. Ethical Considerations

This study follows the fundamental principles of the UNESCO Universal Declaration of Human Rights, the Helsinki Declaration, Spanish Organic Law 3/2018 of 5 December on the Protection of Personal Data and Guarantee of Digital Rights, keeping it strictly confidential and not accessible to unauthorized third parties, and Regulation (EU) 2016/679 of the European Parliament and Council of 27 April 2016 on Data Protection (RGPD) of the Spanish State. The study was approved by the institutional ethical committees from both primary care health zones (Comité de Ética del Complejo hospitalario de Toledo CEITO (273/18) and Comité de Ética e Investigación Provincial de Málaga CEIMA (25/2018)). 

## 3. Results

### 3.1. Clinical and Sociodemographic Variables

A total of 390 women met the participation criteria and completed the online questionnaires. The case group included 152 (39.97%) of these women. Mean BMI was 23.54 ± 4.09, and mean age, 35.49 ± 9.41. The most common etiology of hypothyroidism was autoimmune (*n =* 134, 88.2%). Mean TSH level was 2.22 ± 0.90. In the total sample, 97 (24.9%) women were habitual smokers. There were 292 (74.9%) women with university studies; 285 (73.05%) women were employed. Regarding sexual relations, 349 (89.5%) women were heterosexual, 307 (78.7%) were married or had a steady partner, and 232 (59.5%) were cohabitating (Table 1).

Both groups were compared using Student’s *t*-test (quantitative) and a χ-square test (categorical) (Table 1).

### 3.2. SF-12v1 Results

Mean scores in the case group were significantly lower than in the control group in both summary components (*p <* 0.001; Table 2).

SF-12v1 scores were dichotomized based on the Spanish mean (50 ± 10). Scores below 50 in any of the components show a quality of life lower than the mean; scores above 50 show a quality of life higher than the mean.

In the case group, 39.5% of women scored below the Spanish mean in the PCS (17% in the control group) and 67.75% of them in the MCS (49.5% in the control group). These differences were also significant in both components (*p <* 0.001; Table 2).

Logistic regression was used in order to assess the effect of the independent variables on HRQOL. The variable HRQOL was dichotomized (>50, <50). Regarding the MCS, only hypothyroidism showed a significant influence on HRQOL (*p* = 0.020, OR: 1.98 (1.11–3.52)). Regarding the PCS, a significant influence of BMI (*p* = 0.008) and age (*p* = 0.032) on the total sample and of age (*p* = 0.001) in hypothyroid women (Table 3) was found. All other variables (e.g., etiology, tobacco consumption, civil status) were not significant.

Hypothyroidism increased the risk of a lower HRQOL (<50) in the MCS (*p* = 0.020, OR: 1.98 (1.11–3.52)) and in the PCS (*p* = 0.001, OR: 2.76 (1.47–5.15); Table 3 and Table 4).

The specific influence of hypothyroidism on the PCS was evaluated using an analysis of covariance (ANCOVA), adjusted for “Age” (Model 0) and “Age and BMI” (Model 1) variables. Hypothyroidism remained significant (*p <* 0.001) in both models (Table 4).

## 4. Discussion

HRQOL is a construct widely used in current research to measure subjective well-being. This measurement can be influenced by different sociodemographic aspects such as age, health behavior, physical or mental disease, and productivity [37]. 

In our study, we found that hypothyroid women under LT4 treatment showed significantly lower scores in SF-12v1 physical and mental components than the general Spanish population [33]. Moreover, the proportion of women with HRQOL scores below the Spanish mean was significantly higher in hypothyroid women in both components. These differences could not be explained by other sociodemographic factors (e.g., educational level, etiology, civil status). Age and BMI affected the quality of life; however, after isolating the effect of both factors, hypothyroidism still showed a significant influence on HRQOL. In this sense, our results coincide with those of Djurovic et al. [38], who observed impairment in general well-being in patients with Hashimoto’s thyroiditis. 

Regarding age, Djurovic et al. found significant differences when patients were older than 50 years but not when they were younger than 50 years [38]. However, in our study, hypothyroidism still showed an influence on HRQOL, independent of age. Winther et al. also reported that although HRQOL improved with treatment, it was still worse than in controls after six months of treatment [22]. Unlike our study, they found significant differences only in the SF-36 mental component and not in the physical one [22]. Given that their sample was of older age, physical problems associated with hypothyroidism could resemble those of aging and, therefore, become less evident.

Saravanan et al. reported lower mental well-being in hypothyroid patients despite treatment. Saravanan used a large sample, although, according to the author, with a high proportion of chronic diseases [39]. As noted before, these diseases may hide symptoms of hypothyroidism.

Our data show that HRQOL in levothyroxine-treated hypothyroid women is worse than in controls. Other studies also found a worse HRQOL in hypothyroid patients [19,40,41], although they used reference values for the general population instead of control groups. On the other hand, with a smaller sample, Rakhshan et al. observed no difference in the quality of life between patients with hypothyroidism and a healthy population [32]; however, they reported lower scores in mental health. None of the mentioned studies was sex-specific—all included men in their samples [19,22,32,38,39,40,41]. Therefore, there could be differences between sexes that may explain the differences with our results.

Age and BMI showed an influence on HRQOL in the physical component summary. Aging is related to changes in couple relationships, physical appearance, or sexuality, as well as to climacteric symptoms [42]. On the one hand, climacteric and menopause trigger a decline in the quality of life in women [43,44]; on the other hand, a previous study reported that hypothyroid women showed a higher prevalence of sexual disorder than non-hypothyroid women [11]. It is noteworthy that the mean age of our sample was lower than in other studies about HRQOL and hypothyroidism [31,38,41,45].

Hypothyroid women usually gain more weight than the general population, even after reaching euthyroidism [46]. In our study, women with hypothyroidism had a higher BMI than control women (24.41 ± 4.59 vs. 22.99 ± 3.65). The increase in BMI in treated hypothyroid women has also been reported by other studies [40,47]. However, Singh et al. and Massolt et al. found no differences in the BMI values, although Singh did not have a control group [17,45]. Overweight has been generally associated with a decrease in HRQOL [48]. It also has been related to several health problems, such as hypertension, diabetes, or cardiovascular diseases [49]. In a study with hypothyroid patients under treatment, the authors reported that the increase in BMI was related to decreased QOL [40]. Nonetheless, Michaelsson et al. found no relationship between BMI and quality of life [31].

Hypothyroid patients tend to complain about weight gain due to the disease. These complaints refer to unsatisfactory weight gain after starting the treatment as well as to its impact on psychological well-being and body image, especially in women [39].

In our study, the effects of age and BMI were adjusted through ANCOVA, and it was found that hypothyroidism still showed a significant influence on HRQOL. Therefore, the differences in HRQOL could be explained by the hypothyroidism itself. 

HRQOL may be affected by social, economic, and cultural aspects [37]. Although in our study, different variables were assessed, such as tobacco consumption, educational level, employment status, sexual orientation, civil status, or cohabitation, we found no relation with HRQOL in hypothyroid women. In line with Morón et al. [41], our results showed no relation between HRQOL and tobacco consumption, living arrangement, or educational level. Rakhshan et al. observed a significant relationship between educational level and mental health, but it disappeared when assessing its relationship with quality of life. This author reported no changes in QOL regarding marital status or job status [32]. Regarding etiology, it has been suggested that an autoimmune origin could affect HRQOL [18]. In our study, etiology showed no significant effect on HRQOL.

As previously stated, LT4 allows hypothyroid patients to reach euthyroidism. Nevertheless, this seems insufficient to resolve the symptoms in all the patients. Although its causes are not fully known, different explanations have been proposed. Firstly, the probability of a genetic susceptibility due to the presence of polymorphisms of the type 2 deiodinase enzyme, responsible for the conversion of T4 to T3 in different tissues, has been postulated [12,50]. These alterations could affect the enzyme function and, thus, be related to the persistence of symptoms [12,50,51,52]. Furthermore, it has been suggested that monotherapy with LT4 may not be sufficient to resolve the symptoms of hypothyroidism since it does not replicate the normal physiology of thyroid hormones [18]. Although 80% of T3 is obtained by transforming T4 in T3 through the action of deiodinases, the remaining 20% is directly synthetized by the thyroid gland. The T3 released by the thyroid gland could conduct some direct physiological action [18]. 

It has also been considered that TSH levels may not be the most appropriate marker to assess euthyroidism. Treated patients with normalized TSH levels have been found to have higher levels of T4 and fT4 than controls, higher T4/T3 or FT4/FT3 ratios [53,54,55], and lower serum T3 and FT3 concentrations [47]. Another alternative treatment to LT4 monotherapy is the use of desiccated thyroid extracts (DET). Peterson et al. reported that patients treated with DET showed higher satisfaction with their treatments than patients treated only with LT4. The use of DET causes supraphysiologic T3 serum levels, which may improve mood in patients with depressive symptoms. On the other hand, higher satisfaction could be due to other unknown mechanisms related to thyroid metabolism [56]. In the same way, Mitchell et al. observed that patients treated with DET or an LT3/LT4 combination showed better QOL than patients with LT4 monotherapy. Nevertheless, these differences disappeared when introducing other elements in the analysis (prior healthcare experiences and expectations on treatment) [57]. This study suggested that improving the patient experience and clarifying expectations at diagnosis could improve satisfaction and quality of life [57]. Several studies have researched the effect of the LT3/LT4 combination instead of LT4 monotherapy; however, although some patients prefer the LT3/LT4 combination, the evidence does not conclusively show a clear benefit of this combined therapy [5,18].

Lastly, the influence of other psychological and physical factors has also been considered. Among physical factors, besides the possible influence of BMI, hypothyroidism may be related to the presence of chronic diseases whose symptoms may overlap [18]. In our sample, women had no other severe chronic diseases. Regarding psychological factors, a possible bias related to chronic diseases due to the worse perception of health by these patients must be considered. This effect has been observed in patients with hypertension or diabetes [58]. A higher prevalence of anxiety and depression disorders in these patients has also been reported [50,59], although this might be a consequence rather than a cause of these problems.

### Strengths and Limitations

Among the strengths of this study, the size of its sample is noteworthy as it is larger than in similar previous studies. Moreover, since it was focused on women, the possible presence of gender bias is reduced. To the best of our knowledge, this is the first study comparing HRQOL among hypothyroid women treated with levothyroxine and controls in the Spanish population. On the other hand, this study is not free from limitations. First, as it is a case–control study, it is not possible to establish causal relationships. Second, responses to self-completed questionnaires may be influenced by social desirability.

## 5. Conclusions

Hypothyroid women under levothyroxine treatment showed lower scores in HRQOL than non-hypothyroid women. These differences were significant and could not be explained by any other variable in the study. These lower scores remained even after reaching euthyroidism. Increase in age and BMI were associated with a decrease in the physical component of quality of life, but not in the mental component. An intervention at different levels—assessing the expectations of the treatment, providing complete and appropriate information, nonpharmacological interventions, and research on new treatments—could help improve the HRQOL of these patients. 

## 6. Contributions of This Study

To the best of our knowledge, this is the first study comparing HRQOL among hypothyroid women treated with levothyroxine and controls in the Spanish population.

This study shows that although women with hypothyroidism under levothyroxine treatment may have normalized TSH levels, they still report a low quality of life. More studies of this type are needed, as healthcare providers should be aware of the impact of thyroid disorders on patients’ lives.

## Figures and Tables

**Table 1 jcm-09-03864-t001:** Sociodemographic and clinical variables.

Variable	Case Group	Control Group	Total	*p*
	*n* = 152	*n =* 238	*n =* 390	
	Mean (SD)	Mean (SD)	Mean (SD)	*t*-test
Age	36.58 (9.96)	34.79 (8.99)	35.49 (9.41)	0.068
BMI	24.41 (4.59)	22.99 (3.65)	23.54 (4.09)	0.001
	*n* (%)	*n* (%)	*n* (%)	χ²
Etiology of hypothyroidism				
Autoimmune	134 (88.2%)	-	-	-
Others	18 (11.8%)	-	-	-
Tobacco consumption				
Yes	40 (26.3%)	57 (23.9%)	97 (24.9%)	0.598
No	112 (73.7%)	181 (76.1%)	293 (75.1%)	
Civil status				
Single	24 (15.8%)	34 (14.3%)	58 (14.9%)	0.910
Married/steady	118 (77.6%)	189 (79.4%)	307 (78.7%)	
Other	10 (6.6%)	15 (6.3%)	25 6.4%)	
Sexual orientation				
Lesbian	6 (6.95%)	8 (3.35%)	14 (3.6%)	0.934
Heterosexual	135 (88.82%)	214 (89.95%)	349 (89.5%)	
Bisexual	11 (7.23%)	16 (6.7%)	27 (6.9%)	
Cohabitation				
Yes	101 (66.5%)	131 (55%)	232 (59.5%)	0.025
No	51 (33.5%)	107 (45%)	158 (40.5%)	
Educational level				
Primary	15 (9.85%)	6 (2.5%)	21 (5.4%)	*p <* 0.001
Secondary	39 (25.65%)	38 (15.95%)	77 (19.7%)	
University	98 (64.5%)	194 (81.55%)	292 (74.9%)	
Employment status				
Student	16 (10.5%)	31 (13%)	47 (12.05%)	0.001
Employed	101 (66.5%)	184 (77.3%)	285 (73.05%)	
Other	35 (23%)	23 (9.7%)	58 (14.9%)	

SD: standard deviation.

**Table 2 jcm-09-03864-t002:** HRQOL scores in the case group (women with hypothyroidism) and the control group (women without hypothyroidism).

Variable	Case Group	Control Group	Total	*p*
	*n =* 152	*n =* 238	*n =* 390	*t*-test
PCS	49.64 (10.16)	54.75 (5.76)	52.76 (8.15)	*p <* 0.001
MCS	41.23 (12.12)	46.45 (10.22)	44.41 (11.27)	*p <* 0.001
PCS total score				χ-square
<50	60 (39.5%)	40 (17%)	100 (25.5%)	*p <* 0.001
>50	92 (60.5%)	198 (83%)	290 (74.5%)	
MCS total score				
<50	103 (67.75%)	118 (49.5%)	221 (56.5%)	*p <* 0.001
>50	49 (22.25%)	120 (50.5%)	169 (43.5%)	

PCS: physical component summary; MCS: mental component summary.

**Table 3 jcm-09-03864-t003:** Relationship between independent variables and PCS *.

Variables	Case Group		Control Group		Total Women	
	*p*	OR (CI 95%)	*p*	OR (CI 95%)	*p*	OR (CI 95%)
Hypothyroidism	-	-	-	-	0.001	2.76 (1.47–5.15)
BMI	0.055	0.92 (0.84–1.00)	0.061	0.91 (0.83–1.00)	0.008	0.92 (0.86–0.97)
Age	0.001	0.91 (0.87–0.96)	0.801	1.00 (0.95–1.05)	0.032	0.96 (0.93–0.99)

* Only significant results are shown. BMI: body mass index; OR: odds ratio; CI: confidence interval.

**Table 4 jcm-09-03864-t004:** Mean differences (analysis of covariance (ANCOVA)) in the mental component summary adjusted for age (Model 0) and adjusted for age + BMI (Model 1).

Variables	Case Group		Control Group		*F-test*	*p*
Model 0	M (SD)	CI adjusted	M (SD)	CI adjusted		
PCS	49.82 (0.62)	48.59-51.04	54.64 (0.49)	53.66-55.62	36.56	*p <* 0.001
Model 1	M (SD)	CI adjusted	M (SD)	CI adjusted		
PCS	50.07 (0.61)	48.86-51.29	54.48 (0.49)	53.51-55.44	30.57	*p <* 0.001

M: marginal estimated means ± SD. Model 0: adjusted for age. Model 1: adjusted for age + BMI. Statistical significance (*p* ˂ 0.05) in pairwise mean comparisons using the Bonferroni posthoc test; BMI, body mass index; SD: standard deviation; CI, confidence interval., PCS, physical component summary.

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
