# Peer review of "Health-Related Quality of Life in Levothyroxine-Treated Hypothyroid Women and Women without Hypothyroidism: A Case–Control Study"

_jcm, 2020, doi:10.3390/jcm9123864_

Round 1

Reviewer 1 Report

The authors have compared the health-related quality of life (HRQoL) of a group of hypothyroid patients under treatment with LT4 with a non-hypothyroid control group. HRQoL included both physical component summary (PCS) and mental component summary (MCS). Levothyroxine treatment may improve HRQoL, though there are studies showing no effect of LT4 treatment on HRQoL.

This is a concisely composed study. However, I have some questions and comments:

- Had patients with autoimmune thyroiditis induced hypothyroidism a different HRQoL profile as compared to those without autoimmune disease?

- Did the authors exclude from their study use among patients of supplements such as selenium or vitamins (vit. B12, vit. B6, vit. D3); in the opposite case, did they keep note of their patients’ use of these?

- Multiple parameters, including prior healthcare experiences and expectations, influence patient satisfaction with hypothyroidism treatment and QoL. Enhancing patient experience and clarifying expectations at diagnosis may improve satisfaction and QoL (Mitchell AL et. Clin Endocrinol 2020).

-  Higher satisfaction with both treatment and physicians is reported by those patients treated with

 desiccated thyroid extracts.  While a mechanistic explanation for this observation has not been

  offered, studies should investigate whether preference for this kind of treatment or the addition of

 T3 to LT4 is related to triiodothyronine levels (Peterson S. et al. Thyroid 208).

- The authors should include the above-mentioned studies and briefly comment on them.

- What is the clinical message of this study as related to treatment? Based on these results, should older people with hypothyroidism be treated with levothyroxine?

- The main finding of this study is that hypothyroid women have lower HRQoL than non-hypothyroid patients and this should be more emphatically discussed.

Author Response

The authors have compared the health-related quality of life (HRQoL) of a group of hypothyroid patients under treatment with LT4 with a non-hypothyroid control group. HRQoL included both physical component summary (PCS) and mental component summary (MCS). Levothyroxine treatment may improe HRQoL, though there are studies showing no effect of LT4 treatment on HRQoL.

This is a concisely composed study. However, I have some questions and comments:

- Had patients with autoimmune thyroiditis induced hypothyroidism a different HRQoL profile as compared to those without autoimmune disease?​

Author: Thank you for the suggestions. Scores were compared based on etiology and we found no significant differences. This has been added in the manuscript.

Line 207-208:  All other variables (etiology, tobacco consumption, civil status, etc...) were not significant.

Line 294-295: Regarding etiology, it has been suggested that the autoimmune origin could affect HRQOL [18]. In our study, etiology showed no significant effect on HRQOL.

- Did the authors exclude from their study use among patients of supplements such as selenium or vitamins (vit. B12, vit. B6, vit. D3); in the opposite case, did they keep note of their patients’ use of these?​

Author: Thank you for the question. Besides previous diseases, patients were asked about the medication they were taking. This information allowed us to confirm the absence of diseases or unknown or non-registered treatments. None of the patients reported the use of vitamin supplements.

- Multiple parameters, including prior healthcare experiences and expectations, influence patient satisfaction with hypothyroidism treatment and QoL. Enhancing patient experience and clarifying expectations at diagnosis may improve satisfaction and QoL (Mitchell AL et. Clin Endocrinol 2020).

-  Higher satisfaction with both treatment and physicians is reported by those patients treated with desiccated thyroid extracts.  While a mechanistic explanation for this observation has not been   offered, studies should investigate whether preference for this kind of treatment or the addition of  T3 to LT4 is related to triiodothyronine levels (Peterson S. et al. Thyroid 2018).

- The authors should include the above-mentioned studies and briefly comment on them.

Authors: We honestly appreciate the reviewer’s comments, which have been useful to clarify our manuscript. Both studies have been included and referenced.

Lines 312-326: Another alternative treatment to LT4 monotherapy is the use of desiccated thyroid extracts (DET). Peterson et al. (2018) reported that patients treated with DET showed higher satisfaction with their treatments than patients treated only with LT4. The use of DET causes supra-physiologic T3 serum levels which may improve mood in patients with depressive symptoms. On the other hand, a higher satisfaction could be due to other unknown mechanisms related to thyroid metabolism [56]. In the same way, Mitchell et al. (2020) observed that patients treated with DET or with a LT3/LT4 combination showed better QOL than patients with LT4 monotherapy. Nevertheless, these differences disappeared when introducing other elements in the analysis (prior healthcare experiences and expectations on treatment) [57]. This study suggested that improving the patient experience and clarifying expectations at diagnosis could improve satisfaction and quality of life [57]. Several studies researched the effect of LT3/LT4 combination instead of LT4 monotherapy; however, although some patients prefer the LT3/LT4 combination, evidence does not conclusively show a clear benefit of this combined therapy [5,18].

- What is the clinical message of this study as related to treatment? ​

Authors: As it has been stated in previous studies (Okosieme, O.; Gilbert, J.; Abraham, P.; Boelaert, K.; Dayan,  C.; Gurnell, M.; et al. Management of primary hypothyroidism: statement by the British Thyroid Association Executive Committee. Clin. Endocrinol. (Oxf). 2016, 84, 799-808), levothyroxine is still a safe, simple and effective treatment.

However, it should not be considered that its intake and the subsequent normalization of TSH levels imply the resolution of the symptoms. Although this treatment seems effective, aspects related to different criteria –mood disorders, persistency of the symptoms…– should be assessed. If needed, other actions, not necessarily pharmacological ones, could be added to the treatment in order to improve HRQOL of these patients.

Taking into account this suggestion, conclusions have been enhanced.

Lines 349-356: . These differences were significant and could not be explained by any other variable in the study. These lower scores remained even after reaching euthyroidism. Increase in age and BMI were associated with a decrease in the physical component of quality of life, but not in the mental component. An intervention at different levels –assessing the expectations of the treatment, providing complete and appropriate information, non-pharmacological interventions, and research on new treatments…– could help improve HRQOL of these patients.  

Based on these results, should older people with hypothyroidism be treated with levothyroxine?

Although deciding about such issue was not one of the aims of this study, we found no reasons to interrupt the levothyroxine treatment for any patient. Our goal was assessing if, besides normalizing TSH levels, this treatment improves quality of life. In addition, we aimed at providing with data for future research in order to improve the treatment of this disease.

- The main finding of this study is that hypothyroid women have lower HRQoL than non-hypothyroid patients and this should be more emphatically discussed.

The discussion and conclusions section have been accordingly corrected.

Reviewer 2 Report

The study has been conducted well.

The grammar and sentence formation are inadequate and this should be resolved. I would suggest that the authors should take a relook at improving the presentation of language so their work is better understood.

Author Response

Thank you for your suggestions. English has been revised again by a native speaker.
